# Au-Ag Alloy Nanoshuttle Mediated Surface Plasmon Coupling for Enhanced Fluorescence Imaging

**DOI:** 10.3390/bios12111014

**Published:** 2022-11-13

**Authors:** Kai-Xin Xie, Zhao Li, Jia-Hua Fang, Shuo-Hui Cao, Yao-Qun Li

**Affiliations:** 1College of Chemistry and Materials, Taiyuan Normal University, Jinzhong 030619, China; 2Department of Chemistry and the MOE Key Laboratory of Spectrochemical Analysis & Instrumentation, College of Chemistry and Chemical Engineering, Xiamen University, Xiamen 361005, China; 3Department of Electronic Science, Xiamen University, Xiamen 361005, China

**Keywords:** surface plasmon coupled emission, cell imaging, variable-angle nanoplasmonic fluorescence microscopy, alloy nanoshuttle, fluorescence enhancement

## Abstract

Surface plasmon-coupled emission (SPCE), a novel signal enhancement technology generated by the interactions between surface plasmons and excited fluorophores in close vicinity to metallic film, has shown excellent performance in bioimaging. Variable-angle nanoplasmonic fluorescence microscopy (VANFM), based on an SPCE imaging system, can selectively modulate the imaging depth by controlling the excitation angles. In order to further improve the imaging performance, Au-Ag alloy nanoshuttles were introduced into an Au substrate to mediate the plasmonic properties. Benefiting from the strong localized plasmon effect of the modified SPCE chip, better imaging brightness, signal-to-background ratio and axial resolution for imaging of the cell membrane region were obtained, which fully displays the imaging advantages of SPCE system. Meanwhile, the imaging signal obtained from the critical angle excitation mode was also amplified, which helps to acquire a more visible image of the cell both from near- and far-field in order to comprehensively investigate the cellular interactions.

## 1. Introduction

Fluorescence imaging technology has been extensively applied in chemical and biological research owing to its high sensitivity and specificity [1,2]. A fluorescence enhancement technique, surface plasmon-coupled emission (SPCE) has been successfully used in the field of imaging [3] during which it is referred to as “SPCE imaging”. Benefiting from the directional radiation, distance-dependent coupling, and angular and wavelength resolution of SPCE [4,5], SPCE imaging has displayed excellent performance in bioimaging and biomolecular interaction analysis [6]. SPCE is generated by the coupling interactions between surface plasmons (SPs) and excited fluorophores in close vicinity to a thin metal film (less than 100 nm) during wave–vector matching [7,8,9,10,11]. Several special optical elements have been applied in order to satisfy this matching, such as a high refractive index prism [12,13], high numerical-aperture (NA) objective [14,15], and grating [16,17]. In particular, variable-angle nanoplasmonic fluorescence microscopy (VANFM) [18], a new kind of SPCE microscopy (SPCEM) system based on a high-NA objective, is built on an inverted microscope assembled from a total internal reflection fluorescence (TIRF) illuminator and a polarization controller, which could obtain the detailed sample images of critical angle excitation and surface plasmon resonance (SPR) excitation mode simultaneously during the change of incidence angle. Compared to ordinary fluorescence imaging with interfering signals outside the focal plane, VANFM takes advantages of SPCE to focus the molecular behavior near the surface with improved axial resolution, which has been applied to monitor receptor-mediated endocytosis [18]. Additionally, in view of the different imaging regions of critical angle excitation and SPR excitation mode obtained from VANFM, the basal and lateral membrane domains within a few hundred nanometers of the surface could be discriminated [19]. However, considering the loss of imaging signal induced by the thin coupling region and the complexity of cellular activity, there is still a demand for signal enhancement and further improvement in imaging quality and brightness.

Noble metal nanoparticles (NPs), especially Au and Ag NPs, have received widespread attention owing to their unique localized SPR properties in the visible and near-infrared spectra [20,21,22]. Normally, Au NPs possess higher stability but suffer from weak plasmonic effect due to high Ohmic losses [23], while Ag NPs have a stronger and more sensitive SPR effect but are easily susceptible to oxidation [24]. The stabilizing effect of Au on Ag through alloying has been reported, and NPs with high plasmonic activity and excellent stability could be obtained after alloying [25,26]. To date, Au-Ag alloys, such as plates [27], stars [28], shuttles [29], and islands [30], have been synthesized with tunable SPR properties and have aroused much attention due to their special optical properties for plasmon applications. Among these nanostructures, an Au-Ag nanoshuttle (NS), with an Au nanorod (AuNR) as its core and Au and Ag epitaxial growing to generate sharp tips at both ends, has been employed in enhanced SERS detection [26]. The properties of Au-Ag NSs have several advantages. First, NS possesses strong electromagnetic (EM) field enhancement with sharp tips coupled with a large extinction cross section [31], which could exhibit a strong plasmonic response and mediating ability. Second, under different synthesis conditions, such as the Ag^+^/Au^3+^ ratio, temperature, and pH, the morphology and size of NS could easily be altered to control the plasmonic properties [32,33]. Additionally, the SPR peak is made flexible by changing the aspect ratio of the original AuNR [31], which is conducive to intense interaction with optical waves in order to improve the compatibility and modulation efficiency of NS in plasmon-based optical systems. Moreover, the growth of arrow-headed NS with an Au and Ag alloy shell is a thermodynamically driven process, therefore NS holds a thermodynamically favored shape with stable properties [33,34]. The Au-Ag nanoshuttle possesses a stable structure with high plasmonic activity, which is expected to be a powerful candidate for modulating the optical performance of plasmon-based imaging and sensing platforms.

In this work, we developed an enhanced cell imaging strategy based on Au-Ag NSs modification through electrostatic adsorption employing VANFM. After the introduction of NSs, a strong EM field induced by the localized SPs of NPs was obtained, thereby improving the imaging performance for the ultra-thin fluorophore layer and the near-field region of cell sample, which could further reveal the advantages of SPCE imaging. Additionally, an enhanced imaging signal collected from SPR excitation and critical angle excitation could be acquired at the same time, which could yield a more visible image of the cell both from near- and far-field in order to collect complete information. Moreover, the Au-Ag NS-mediated SPCE imaging is a first attempt to improve the imaging performance, which provides the possibility and basis for the effective mediation of other metallic nanoparticles in the SPCE imaging field.

## 2. Materials and Methods

### 2.1. Materials

Rhodamine B (RhB; Sigma-Aldrich, St. Louis, MO, USA), poly-(methyl methacrylate) (PMMA; MW: 350,000; Alfa Aesar (China), Shanghai, China), anisole (Sigma-Aldrich, St. Louis, Mo, USA), cysteamine (Sigma-Aldrich, St. Louis, MO, USA); Au-Ag alloy nanoshuttle solution (Nanjing NANOEAST Biotech Co., Ltd., Nanjing, China); 1,1′-dioctadecyl-3,3,3′,3′-tetramethylindocarbocyanine perchlorate (DiI; Sigma-Aldrich, St. Louis, MO, USA), paraformaldehyde (PFA; Alfa Aesar (China), Shanghai, China). Ethyl alcohol, dimethylsulfoxide (DMSO), H_2_SO_4_, NaCl, KCl, Na_2_HPO_4_·12H_2_O, and KH_2_PO_4_ were all analytical grade reagents (all from Sinopharm Chemical Reagent Co., Ltd., Shanghai, China). Ultrapure water was prepared using a Milli-Q system. The synthesis process of Au-Ag NSs is mentioned in the supporting information.

### 2.2. Au Substrate Preparation and Modification

Au substrate was prepared by depositing 2 nm Cr and 30 nm Au separately on a clear coverslip (FIS12-545-102; Fisherbrand, Waltham, MA, USA) through magnetron sputtering. The substrate was incubated in 10 mM cysteamine for 1 h, thoroughly rinsed with ethanol, and dried in N_2_. Then, the substrate was immersed in the Au-Ag NS solution for 3 h in order to obtain the NSs-modified chip. The extinction spectrum and XRD patterns for Au-Ag NSs are shown in Appendix A.

### 2.3. Preparation of RhB-PMMA Solution

The RhB-PMMA anisole solution was prepared with a weight percentage of 1%, and the RhB concentration was 0.1 mmol/L.

### 2.4. Cell Culture

Hela cells were cultured in Dulbecco’s modified Eagle’s medium (DMEM; HyClone, Logan, UT, USA) supplemented with 10% fetal bovine serum (HyClone, Logan, UT, USA) at 37 °C under 5% CO_2_. Cells in the logarithmic phase were digested by trypsin and incubated in the cleaned bare substrate and modified substrate overnight for cell adherence.

### 2.5. Chip Preparation for Imaging

For the VANFM of the enhancement model, the RhB-PMMA solution was spin-coated on the modified substrate at 3500 rpm for 30 s and dried in air in order to obtain a dye-doped chip. For the blank experiment, the RhB-PMMA solution was spin-coated on the bare Au substrate without NSs modification in the same conditions.

For the VANFM of cell imaging, the labeling process for RhB was as follows. The modified chip was washed three times with PBS buffer and fixed with a 4% PFA for 30 min. Then, the chip was washed three times with PBS and labeled in RhB/PBS solution (80 μL RhB ethyl alcohol solution with a concentration of 0.1 mmol/L into 2 mL PBS) for 20 min at room temperature. The labeling process for DiI was as follows: the modified chip was washed three times with PBS and labeled in DiI/PBS solution (6 μL DiI DMSO solution with a concentration of 0.1 mg/mL into 2 mL PBS) for 20 min at room temperature. Then, the chip was washed three times with PBS buffer and fixed with a 4% PFA for 30 min. The chips were washed clean with PBS prior to measurement. For the blank experiment, the substrate for cell adherence was bare Au substrate without NSs modification.

### 2.6. Variable-Angle Nanoplasmonic Fluorescence Microscope

Fluorescence images were obtained via a microscope built on the commercial TIRF microscope (Ti-U; Nikon, Tokyo, Japan) with a λ/2 plate inserted in the incident light pathway in order to adjust the incident polarization. The images were taken at different excitation angles with a range from 0° (epifluorescence illumination, EPI) to 80°, and the angular distribution curves of fluorescence were obtained after analyzing the images’ region of interest (ROI). The emission spectrum was collected from the homemade fluorescence detection system which was connected to another emission path of VANFM. For micro-spectrum acquisition, the imaging should be focused on the strongest signal plane according to the excitation angle distribution, and the total internal reflector was adjusted to import the signal into the spectral detector. Then, the spectrum of the corresponding region was collected by monochromator and CCD.

## 3. Results and Discussion

### 3.1. Au-Ag NS Mediated VANFM with RhB-PMMA Layer as a Model

A RhB-PMMA fluorophore layer was employed as the model to investigate the enhancement effect of Au-Ag NS. The schematic is shown in Figure 1A. As mentioned earlier, imaging collected from critical angle excitation (*θ_c_*) and SPR excitation (*θ_sp_*) could be acquired at the same time in VANFM. In the SPR excitation mode, only the p-polarized incident light could excite the SPs and then couple with fluorophores nearby to generate SPCE. The images, spectra, and excitation angular distribution for the RhB-PMMA layer without NSs obtained from the SPR excitation mode using different polarizations are shown in Figure 1C,E (black and red curves) and Appendix A. As shown in these figures, the polarization-dependent excitation (the signal intensity of p-polarized excitation was 20 times that of the s-polarized excitation) and directional excitation (highly directional angle around 50°) were well presented, indicating the effective coupling process of SPs and the fluorophores in close vicinity of the Au substrate. The thickness of the fluorophore layer was about 20 nm, which was confirmed by comparing the data obtained from the experimental emission angle (Figure 1F) and calculated SPR angle (Appendix A). Specifically, the thickness of the fluorophore layer was the same as the datum used for SPR simulation when the SPR angle was equal to the experimental excitation angle of SPCE (50°). By comparing the images in Figure 1B (the EPI imaging of RhB-PMMA layer without NSs) and 1C, the advantage of plasmonic coupling imaging is evident for the thin sample. In this system, the fluorophore was too thin to be detected by the critical angle excitation mode (the detection region is located about 100 to 200 nm from the substrate [35]), while the imaging obtained from SPR excitation (*θ_sp_*, 50°) was manifested, which also proves the advantages of near-field coupling of SPCEM for thin sample imaging.

After the electrostatic adsorption of Au-Ag NSs, the basic properties of SPCE remained the same (Figure 1F), but high image brightness and enhanced emission intensity were achieved by comparing the images (Figure 1C,D) and emission signals (Figure 1E). The imaging performance was improved by the introduction of NS and is helpful for the observation of ultra-thin samples. The TEM image and mapping element analysis of Au-Ag NSs are shown in Appendix A. As shown in the TEM image, AuNR was completely embedded into a homogeneous shell, and an arrow-headed structure was generated at both ends of the NPs (the length was about 125 nm; the widths of the rod and tip portions were about 35 nm and 48 nm, respectively). Additionally, the Au and Ag atoms were uniformly mixed in the shell, which can be proven by the mapping element analysis, and the NS has an anisotropic structure, which can also be proven through the transverse and longitudinal SPR bands located at around 500 nm and 800 nm, respectively (Appendix A). As shown in the finite-difference time-domain (FDTD) calculations for the EM field distribution of Au substrate and single Au-Ag NS (Figure 2), the EM field of the localized SPs around NS is mainly distributed in the tip region, and its strength is obviously stronger than that of the bare Au substrate. The enhancement of the EM field restricted at the tip composed of homogeneous Au and Ag atoms takes the main responsibility for the improved imaging performance produced by NS modification. The coupling efficiency of fluorophore and SPs is sensitive to the distance between them. Normally, good signal reproducibility and high coupling radiation can be obtained with a distance of about 50 nm [36]. The matching axial size of NS after modification on Au substrate is beneficial for better enhancement performance. The reproducibility of the enhancement for this model system by Au-Ag NS was investigated, and the related spectra and enhancement factors are shown in Appendix A. It can be seen that the reproducibility of the NS-enhanced system is excellent, which lays a foundation for the stable enhancement of cell imaging by NS.

The concentration of Au-Ag NS for modification was optimized in order to achieve better enhancement. The normalized intensities for different dilution times (10 times to 600 times) are shown in Appendix A. For the high modification concentration, the aggregated NPs are visible (Appendix A), and the fluorescence imaging is greatly influenced by the aggregation which could change the structure and property of NSs. Additionally, the aggregation of NPs will lead to an inhomogeneous coating of fluorophore layer, which could also weaken the emission signal. The images, related fluorescence spectra, and excitation angular distribution for the range of dilutions from 400 to 600 times are shown in Figure 3. The signal first increased and then decreased with the gradually reducing concentration. These results indicate that the optimal NSs solution was produced at 500 dilutions of the original, which was employed to fabricate substrate for the imaging of Hela cell.

### 3.2. The Cell Imaging Enhancement by Au-Ag NS through VANFM

The RhB was further used to stain Hela cells in order to investigate the enhancement performance of Au-Ag NS. The RhB dyeing of cells was nonselective, and the whole cell was stained. Only the observable excitation angle of *θ_sp_* was obtained when the VANFM instrument was used to image a thin fluorophore layer considering the near-field coupling process of SPCE (Figure 1F). However, for Hela cell imaging, emissions from both *θ_c_* and *θ_sp_* excitation appeared. The normalized intensity of the RhB-dyed cell with different excitation angles is shown in Figure 4A (red curve), and the signal peaks appeared at about 60° and 75°. At the incident angle of *θ_c_* (about 60°), evanescent field excitation exists, and an enhanced fluorescence signal can be obtained. When the p-polarized excitation light is incident at the angle satisfying the SPR condition (*θ_sp_*, about 75°), the fluorophores located in the near-field range can be excited and then coupled with SPs to produce an enhanced fluorescence signal. The images in Appendix A and the curves in Figure 4A (black and red curves) demonstrated the polarization-dependent excitation of SPCEM. The images obtained from the critical angle excitation mode mainly included information on the cytoplasmic region away from the substrate within a distance between 100 nm and 200 nm (Appendix A), while the SPCE imaging mainly targeted the membrane near the Au substrate, which demonstrates the background suppression of SPCE (Figure 4B). Though it is roughly recognizable for the regional structure, the images of outlines and uneven parts are smeared. After modification with Au-Ag NS, the imaging brightness and emission intensity were enhanced to increase clarity (Figure 4A,C), and the whole morphology and contour of the cell membrane can be displayed with better imaging quality. The signal-to-background ratio (SBR) for the imaging of RhB-dyed cells with and without NSs is shown in Appendix A through analysis of the ROI of the background and cell image, and the increased SBR verifies the improvement of image quality induced by NSs modulation. In this way, the imaging of the near-field region can be clearly observed for thick samples through the background suppression of SPCE.

Compared with the images generated by critical angle excitation, SPCEM possesses a much thinner imaging region in the proximity of metal substrate due to the distance-dependent coupling of SPCE. Both theoretical and experimental studies have shown that the detection thickness can be limited to around 50 nm [36,37]. In order to better illustrate the advantages of SPCEM imaging, the cells with DiI dyeing were used to study the enhancement of Au-Ag NS. For DiI-stained cells, the coupling area near the Au substrate, that is to say, the cell membrane region, tends to be stained. As shown in Figure 5A, the fluorescence intensities of images obtained from the excitation of 60° and 75° were peak, which corresponded to *θ_c_* and *θ_sp_* excitation, and the result was the same as that of the excitation angle distribution of RhB dyeing. The normalized fluorescence intensity as a function of the incident angle of DiI-dyed cells without NSs is shown in Figure 5D (red curve), and the results also supported this statement. However, the intensity distribution of these two peaks was shown to be quite different when comparing the curves in Figure 4A and Figure 5D. For the cell imaging conducted with RhB total staining, the fluorescence signal excited at *θ_c_* was stronger, since the cell regions away from the substrate were also stained, and the far-field fluorescence emission within critical angle excitation mode can be collected with high efficiency (Appendix A). In contrast, the fluorescence imaging obtained from *θ_sp_* excitation was slightly weaker, which was caused by the thin staining region for coupling. For cell imaging with DiI staining, a stronger imaging signal can be excited by the SPR mode by contrast with the critical angle mode (Figure 5D, red curve), as the staining was mainly directed at the cell membrane region, and the imaging under the SPR excitation mode is clearer as a result of effective coupling of SPCE (Figure 5A).

After the electrostatic adsorption of Au-Ag NSs on the Au substrate, the measured images, intensity, and excitation angular distribution are shown in Figure 5B–E. First, the quality of imaging generated by *θ_sp_* excitation (75°) was improved significantly, which reflects that the modification of NSs has a better enhancement effect in the near-field around the substrate and NPs. Along with a more than three-fold enhanced signal, the SBR of the image was also improved more than two-fold (Figure 5E and Appendix A). Basically, SPCE imaging could obtain a high-resolution image of the near-field area of the substrate; however, this was accompanied by a loss of signal strength when considering the imaging area only for a staining sample with a thickness of tens of nanometers. This statement could also be proven by the comparison between the SPCE imaging obtained from the SPR excitation mode, as shown in Figure 5A (75°), and the EPI membrane image of DiI-dyed cells on a coverslip, as shown in Appendix A. As shown in Appendix A, the cell edge was bright and the inner area was indistinct, which indicates that the imaging involved a certain axial thickness and that the fluorescence information beyond the focal plane was also collected; these phenomena, as well as the existence of an optical diffraction limit, all suggest that ordinary fluorescence imaging could not truly target the membrane area. In contrast, the morphology of the cell membrane was clear with more details, such as the bulges and contour, and the intensity distribution for the different regions of the cell was relatively uniform, which implies that SPCEM was applied to the imaging of a thinner membrane area but was suffering from a weak signal (Figure 5A). After the enhancement of NSs, imaging with high SBR and strong signal was achieved, which makes SPCEM a powerful observation tool for the behavior of cell membrane regions. Second, as shown in Figure 5B (60°), the contour of the cell was very bright, which was caused by the axial accumulation of stained cell membrane, because the curvature of the cell membrane is higher at the cell edge, and the imaging obtained from excitation of *θ_c_* (60°) targets deeper cellular regions. Nevertheless, for the SPR excitation mode, what is shown was a flat and uniform image of thin membrane attached to the Au substrate (Figure 5B, 75°), which represented the difference in imaging performance of critical angle excitation and SPCEM. With the increase of excitation angle, the image intensity at the edge of the cell gradually decreased. Meanwhile, the intensity change trend of the cell center was the opposite, because the imaging plane became shallow and finally reached the vicinity of the metal surface, which displays the axial resolution property of SPCEM. Compared to Figure 5A, a higher axial resolution was achieved, thereby yielding a detailed view of the membrane region after the introduction of NSs (Figure 5B). Third, the property of excitation polarization matching was pretty obvious, and the peak at *θ_sp_* (75°) could be collected only within p-polarized excitation (Figure 5B,C). Moreover, as shown in Figure 5D (red and blue curves) and the images in Figure 5A,B excited at *θ_c_* (60°), the fluorescence intensity and imaging brightness were also enhanced for the critical angle excitation mode, which mainly collects imaging information from the far field. Profiting from the imaging performance improvement for both near- and far-field regions induced by NSs modification, more complete information could be obtained about the cell sample. Additionally, considering the influence of fluorescence bleaching on imaging, the signal enhancement helps to achieve long-term observation of complex cell behavior.

## 4. Conclusions

Benefiting from the enhancement of the EM field brought about by localized SPs, the quality and SBR of images of ultra-thin samples and dyed cells were improved by employing the Au-Ag NSs-modified SPCE chip. The aim of developing a universal strategy for enhancing the SPCEM imaging performance and axial resolution was achieved through a simple electrostatic adsorption modification method. The imaging of the ultra-thin fluorophore layer and the near-field region of complex cell samples can be clearly observed, which benefits from the advantages of SPCE imaging, and this is promising for the observation of cellular membrane-related complex life behavior combined with the background suppression of SPCE, such as cell proliferation and migration. Additionally, the intensity distribution for SPR excitation and critical angle excitation varied for different dyeing regions, which further shows the near-field coupling of SPCE and the imaging advantages for VANFM system. Moreover, the performance of the VANFM system is improved due to the simultaneous enhancement of the near- and far-field imaging brightness, offering an opportunity for long-term cell cultivation, observation, and comprehensive cellular analysis, which is beneficial for the investigation of cellular interactions in situ.

## Figures and Tables

**Figure 1 biosensors-12-01014-f001:**
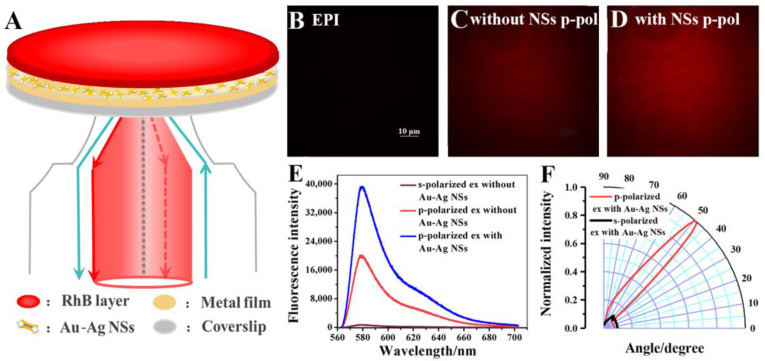
(**A**) The schematic for the Au-Ag NS enhanced RhB-PMMA fluorophore layer imaging. (**B**) The EPI image of fluorophore layer. The images of fluorophore layer (**C**) without and (**D**) with Au-Ag NSs modification excited by p-polarized incident light. (**E**) The SPCE spectra of the fluorophore layer without and with Au-Ag NSs enhancement. (**F**) The excitation angular distribution of the fluorophore layer with Au-Ag NSs modification.

**Figure 2 biosensors-12-01014-f002:**
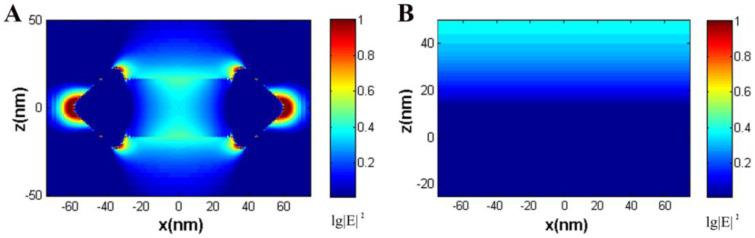
FDTD simulation results of the near field at 561 nm for (**A**) a single Au-Ag NS and (**B**) bare Au substrate.

**Figure 3 biosensors-12-01014-f003:**
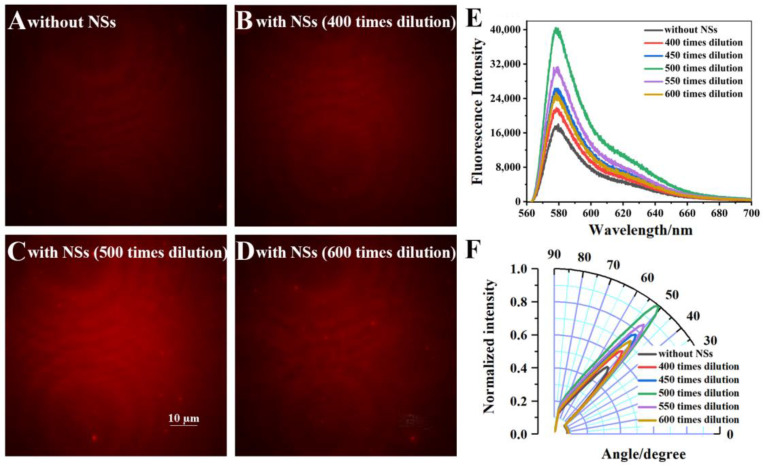
Images of the RhB-PMMA fluorophore layer (**A**) without Au-Ag NS and with different concentrations of Au-Ag NS [(**B**) 400 times, (**C**) 500 times, and (**D**) 600 times dilution]. (**E**) The fluorescence spectrum and (**F**) excitation angular distribution for RhB-PMMA fluorophore layer with different concentrations of Au-Ag NS.

**Figure 4 biosensors-12-01014-f004:**
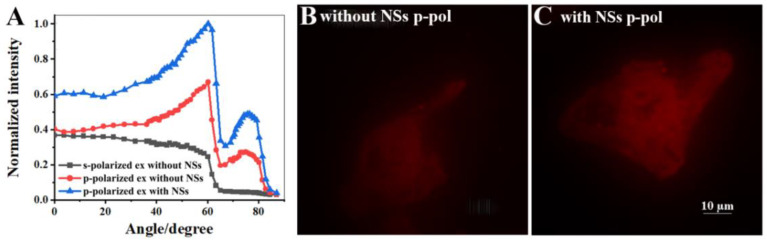
(**A**) The relationship between the normalized intensity of RhB-dyed cell images and the excitation angle. The RhB-dyed cell images from SPR excitation (**B**) without and (**C**) with Au-Ag NSs modification.

**Figure 5 biosensors-12-01014-f005:**
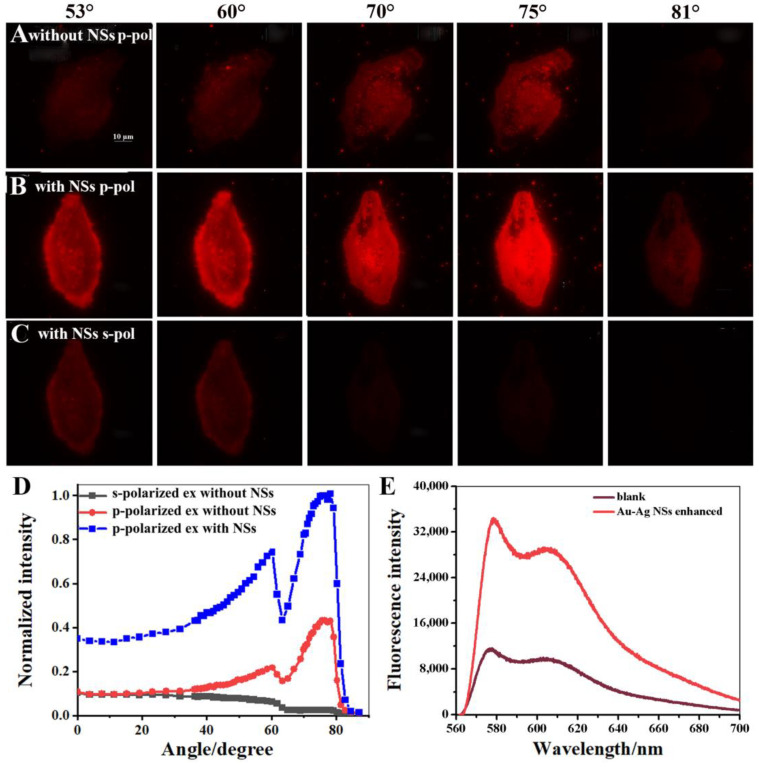
(**A**) The images of DiI-dyed cells at different excitation angles without Au-Ag NS modified on Au substrate. The images of DiI-dyed cells at different angles of excitation with (**B**) p-polarization and (**C**) s-polarization of the Au-Ag NS enhancement system. (**D**) The relationship between the normalized intensity of DiI-dyed cell images and excitation angle. (**E**) The fluorescence spectra of the DiI-dyed cells by SPR excitation with and without Au-Ag NS enhancement.

## Data Availability

Not applicable.

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
