# Peer review of "Au-Ag Alloy Nanoshuttle Mediated Surface Plasmon Coupling for Enhanced Fluorescence Imaging"

_biosensors, 2022, doi:10.3390/bios12111014_

Round 1
Reviewer 1 Report
Authors done good work in drafting the paper. There are few comments below.
1. Did author try extinction spectra of Au-Ag nanoparticles with different Au:Ag ratios?
2. Did author try XRD pattern analysis for Au-Ag nanoparticles with different Au:Ag ratios?
3. Page 3, line 147, change Figure D to Figure 1D.
4. Page 3, line 146, change Figures 1E to Figure 1E.
5. Page 3, line 146, change Figures 1F to Figure 1F.
6. Page 5, line 197, change Figures C to Figures S5C.
7. Page 7. line 242, change Figures 5B to 5E.
Reviewer 2 Report
The article is well written, this is an application of Ag/Au alloy nanoshuttle for fluorescence imagine. The nanoshuttle is buyed by an external company, in my opnion this is a big bug of the manuscript, the center key of the chip is the nanoshuttle, that is not sinthesized by the authors. The author are invited to add informations about the nanomaterial: how were the shuttles sinthesized? what is the solvent? why nanoshuttle and not other shapes? How is the UV-Vis spectrum? Also what are the advantages of this materials with respect to the others reported in literature? The rest of the manuscript is well explained. In the introduction, several manuscript about Ag and Au nanomaterials have been omitted, please add to the introduction: Biosensors 2019, 9(2), 78; https://doi.org/10.3390/bios9020078; - Scientific Reportsvolume 9, Article number: 9082 (2019), https://doi.org/10.1038/s41598-019-45689-9 -Coatings 2020, 10(3), 288; https://doi.org/10.3390/coatings10030288
Reviewer 3 Report
The manuscript ID biosensors-1974294 mainly presents a study about particular the Au/Ag alloy nanostructures incorporated in Au substrates with potential applications for fluorescence imaging assisted by plasmonics. A list of comments for the authors is below:
1. The justification behind the selection of Au/Ag nanoshuttles (Au nanorod as core with Au and Ag epitaxial growing) instead of other hybrid shapes like stars, islands or prisms is not clear. Better details should be provided.
2. If possible, please substitute fluorescence data by quantum yield data in order to better compare the value of the main findings considering other advanced materials.
3. The importance and influence of the size of the samples over the main findings must be described. It is not clear how were determined the particular size of the nanoparticles and film for this particular application.
4. The plasmonic response of samples of Au and Ag is not reciprocal to the plasmonic response of Ag and then Au. Is there any possible perspective with this consideration for proposing future work? The authors are invited to see for instance: doi:10.1088/0957-4484/22/35/355710 and discuss.
5. The Uv-vis absorption spectrum of a representative sample of nanoshuttles would be welcome.
6. Error bar in experimental data must be provided.
7. Please comment about the reproducibility and statistics in the measurements reported.
8. My main concern is the absence of confrontation of advantages and disadvantages of this work with updated publications in the same topic. You can see for instance: https://doi.org/10.1021/acsabm.1c00320
9. How is the evolution of the samples for the enhanced fluorescence imaging? Does the Ag is not sensitive to the oxygen for considering real applications with samples that are not used as-prepared?
10. Some of the collective citations in this work should be split in order to better present the importance of each reference in the description of the topic. Expressions together to the correspondent individual citation could improve the presentation.
Round 2
Reviewer 2 Report
The authors improved positively the quality and scientific soundness of the article.
The manuscript can be published in this form.
Author Response
We are grateful to the reviewer for the efforts to improve our paper.
Reviewer 3 Report
The authors have clarified fundamental points raised I the review stage; however, important issues are still present, please see below:
4. The plasmonic response of samples of Au and Ag is not reciprocal to the plasmonic response of Ag and then Au. The discussion about this issue in the manuscript neither in the reply did not describe details about the coupling claimed in the title; and what this study adds to consider more attractive the configuration proposed for future applications.
5. The bands illustrated in the Uv-vis absorption spectrum of a representative sample of nanoshuttles are not in the typical position of Ag and Au. The authors are invited to explain the differences or correct in case a calibration of the measurement is required.
Author Response
Response to the reviewer’s comments:
The authors have clarified fundamental points raised I the review stage; however, important issues are still present, please see below:
- The plasmonic response of samples of Au and Ag is not reciprocal to the plasmonic response of Ag and then Au. The discussion about this issue in the manuscript neither in the reply did not describe details about the coupling claimed in the title; and what this study adds to consider more attractive the configuration proposed for future applications.
We appreciate the reviewer’s comments. We apologize for the unclear expression of this alloy nanoparticle structure by the representation of Au/Ag in the title. Actually, in this alloy structure, Au and Ag elements have been mixed with each other in the shell part. As shown in Figure R1, the mapping elements analysis demonstrated that Au and Ag elements are homogeneously distributed on the shuttle shell. This structure has also been proved in the literature (ACS Appl. Mater. Interfaces 2014, 6, 3331-3340; small 2015, 11, 5214-5221). Therefore, the Au and Ag sequential plasmonic effect should not happen in our study. The FDTD simulation shown in Figure 2A was for the nanoshuttle with homogeneous Au-Ag alloy shell, which shows a markedly stronger electromagnetic field than that of bare Au substrate. The enhanced coupling efficiency between fluorophores and electromagnetic field mediated by Au-Ag alloy nanoshuttle takes the main responsibility for the improved SPCE imaging performance, and this improvement is a result of the combined plasmonic action of Au and Ag. The mapping analysis for Au and Ag has been added in the supporting information (Figure 4), and the description of this structure has been added in the second paragraph of introduction and the second paragraph of results and discussion. The representation of “Au/Ag” in the nanoshuttle has been changed to “Au-Ag” in the manuscript.
Au-Ag NS possesses a strong electromagnetic (EM) field enhancement with sharp tips, coupled with the large extinction cross section (Opt. Express 2008, 16, 14288-14293), which could exhibit a strong plasmonic response and mediating ability. Besides, under different synthesis conditions, such as Ag to Au ratio, temperature and pH, the morphology and size of NS could be easily altered to control the plasmonic properties (Appl. Surf. Sci. 2020, 528, 146935; ACS Appl. Mater. Inter. 2014, 6, 3331-3340), and the SPR peak is flexible by changing the aspect ratio of the original AuNR (Opt. Express 2008, 16, 14288-14293), which is conducive to the intense interaction with optical wave to improve the compatibility and modulation efficiency of NS in the plasmon-based optical systems. Moreover, the growth of arrow-headed NS with homogeneous Au and Ag alloy shell is a thermodynamically driven process, so that NS holds a thermodynamically favored shape with stable properties (J. Phys. Chem. C 2008, 112, 3203-3208). The Au-Ag alloy nanoshuttle possesses a stable structure with high plasmonic activity, which expects to be a powerful candidate for modulating the optical performance of plasmon-based imaging and sensing platforms. The description of these advantages has been revised and shown in the second paragraph of introduction.
Figure R1. The EDX mapping under TEM observation of Au-Ag alloy nanoshuttle
- The bands illustrated in the Uv-vis absorption spectrum of a representative sample of nanoshuttles are not in the typical position of Ag and Au. The authors are invited to explain the differences or correct in case a calibration of the measurement is required.
Thanks for the reviewer’s suggestion. The positions of UV-Vis absorption peaks are related to the material, morphology and size of nanoparticles. The absorption position and number are different from the traditional Au and Ag sphere nanoparticle (a single band around 400 nm for Ag nanosphere and a single band around 500 nm for Au nanosphere) because of the anisotropic structure of nanoshuttle. Au-Ag alloy nanoshuttle exhibits both transverse and longitudinal surface plasmon resonance (SPR) bands. One weak band in the short wavelength region is assigned to the transverse SPR (TSPR, consistent with the SPR band of the spherical nanoparticles), and the other strong band in the long wavelength region is contributed by a longitudinal SPR (LSPR), which is sensitive to the aspect ratio (AR, the length divided by the diameter) (Opt. Express 2008, 16, 14228; ACS Appl. Mater. Interfaces 2014, 6, 3331-3340; J. Phys. Chem. C 2008, 112, 3203-3208). The two peaks appeared at around 500 nm and 800nm in the UV-Vis absorption spectrum of the alloy nanoshuttle used in our manuscript (Figure R2) were consistent with the analysis and literature reported (Opt. Express 2008, 16, 14228; ACS Appl. Mater. Interfaces 2014, 6, 3331-3340). The description of these two absorption peaks has been added in the second paragraph of results and discussion.
Figure R2. The extinction spectrum for Au-Ag alloy nanoshuttle
